https://doi.org/10.5194/wes-2025-193 Preprint. Discussion started: 15 October 2025

© Author(s) 2025. CC BY 4.0 License.

# Trimming a fixed-wing airborne wind system for coordinated circular flights

Duc H. Nguyen<sup>1</sup>, Mark H. Lowenberg<sup>1</sup>, and Espen Oland<sup>2</sup>

<sup>1</sup>School of Civil, Aerospace, and Design Engineering, University of Bristol, Bristol, BS8 1TR, United Kingdom <sup>2</sup>Kitemill AS, Vanse, 4560, Norway

Correspondence to: Duc H. Nguyen (duc.nguyen@bristol.ac.uk)

Abstract. Airborne wind energy systems (AWES) are tethered flying devices used for electricity generation. During the power generation phase, the aerial component usually flies in a circular or figure-of-eight pattern. This paper examines the control surface movements required for circular flights in fixed-wing AWES. In the absence of gravity, steady trim with equilibrium solutions can be achieved if the orbit plane is normal to the wind. The radius depends on how much the aircraft leans into the turn: leaning in reduces the radius and is statically stable, while leaning out achieves a larger radius but is unstable. For the latter case, artificial stabilisation can be done by cross-feeding the pitch and roll responses to the aileron. For circular trajectories that are not normal to the wind (i.e., experiencing out-of-plane wind), energy needs to be added to the system through periodic forcing of a control surface. Correct timing of the forcing will excite the orbit's natural frequency, enabling full control of the circle centre and orientation for navigation in 3D space. This can be done even in the presence of gravity, which is discussed in the second half of this paper. The aileron is the most effective control effector for forcing. Although the trimming method presented in this paper is only suitable for theoretical studies, it provides insights into the flight dynamics of fixed-wing AWES and lays the groundwork for future flight control developments.

## 1 Introduction

25

Airborne wind energy systems (AWES) harvest wind energy using a tethered flying device. The main advantage of AWES is its lightweight construction, which reduces material usage and lifetime carbon footprint compared with conventional wind turbines of similar outputs. Research into AWES has grown significantly in the past two decades, with research groups and startups proposing increasingly innovative designs. An overview of AWES technology can be found in (Pereira and Sousa, 2023).

Since AWES lack a solid structural foundation seen in conventional wind turbines, sophisticated control strategies are needed to ensure safe and efficient operations. AWES control is a large topic and can be split into flight control and ground winch control, with the former further split into distinct phases of operation (launch, power generation, retraction, and landing). In this work, we focus on the power generation phase of ground-gen AWES, which generates energy by converting kinetic energy

into traction force on the tether which is connected to the ground generator. The flight path in the power generation phase is usually in figure-of-eight or circular pattern (Eijkelhof et al., 2024). Circular flight is the focus of this paper (see Figure 1). In this work, we refer to the tethered aerial component as an aircraft instead of a kite to underline the use of traditional control surfaces (aileron, elevator, and rudder).

Figure 1: a full power production cycle, showing the power generation phase (flown in circular pattern) followed by retraction.

Illustration by Kevin Yu at the University of Bristol.

Various flight control strategies for the power generation phase have been presented, such as those by (Rapp et al., 2019), (Trevisi et al., 2022), and (Fernandes et al., 2022). To corroborate the understanding of those AWES control laws, this paper provides an in-depth discussion of how tethered circular flight is achieved in fixed-wing AWES. The analysis is done in an 'open loop' manner to determine which control surface movements are required. As such, the results do not represent a feedback control implementation, but provide a deeper understanding of AWES flight mechanics and can inform the design of feedback control laws.

Prior work have successfully analysed AWES flight as a static problem (Trevisi et al., 2021) and (Rapp, 2021). In this work, we expand the discussion to examine the cyclic nature of AWES flight due to the presence of gravity and out-of-plane wind. It will be shown that AWES circular flight requires periodically exciting one of the three control surfaces at the natural frequency of the cycle, with the aileron being the most effective option. The use of aileron agrees with observations from a prior study, which noted that the most simple control strategy for AWES is to cyclically actuate the aileron – see chapter 7 in (Trevisi, 2024). Our analysis on the cyclic control inputs takes a time-domain-based approach, which provides an alternative perspective to the frequency-domain analysis provided in (Trevisi et al., 2022).

The first result section of this paper focuses on static trim in zero gravity, noting the difference in static stability between small and large radius circular orbits. A method to stabilise the unstable large-radius orbits is then proposed. Subsequently, dynamic trim is introduced to achieve circular flight in the presence of out-of-plane wind and gravity. The dynamic trim method will be shown to be a means of exciting the natural frequency of the circular orbit trajectory, which provides the energy needed to achieve circular flight despite the energy losses from out-of-plane wind and gravity.

## 2 Simulation model

The analysis was done using a 6-degree-of-freedom simulation of Kitemill's KM1 AWES. The KM1 is a 20 kW groundgen prototype driven by a 54 kg, 7.4 m wingspan aircraft with vertical takeoff and landing capability (see Figure 2). The simulation contains the states listed in Table I. Previous studies that have used the same simulation include (Mohammed et al., 2024b; Mohammed et al., 2024a). CFD was used to construct the aerodynamic data. For illustrative purposes, Figure 3 and Figure 4 present a small set of key aerodynamic relationships as functions of angle of attack and sideslip. The aircraft also has a set of flaps for additional lift during power production, which is set to its maximum deflection of 13° in this paper.

Figure 2: Aerial component of Kitemill KM1 system.

Table I. Summary of the states in the simulation

| Element      | Number of            | Description                               |
|--------------|----------------------|-------------------------------------------|
|              | states               |                                           |
| Aircraft     | 12                   | Body-axis velocities $u, v, w$            |
|              |                      | Boxy-axis angular rates $p, q, r$         |
|              |                      | Earth-axis Euler angles $\phi,	heta,\psi$ |
|              |                      | Earth-axis coordinates $X, Y, Z$          |
| Tether nodes | 6N                   | Each node has 6 states: its Earth-axis    |
|              |                      | coordinates and velocities. All results   |
|              | <i>N</i> : number of | use $N = 15$ , except for Figure 5, which |
|              | tether nodes         | is for illustration only.                 |




Figure 3: Force (a) and moment (b) coefficients as functions of angle of attack at zero sideslip, flaps, and control surface deflections.

Figure 4: Force (a) and moment (b) coefficients as functions of sideslip angle at 5° angle of attack, zero flaps, and no control surface deflections.

The tether model accounts for both tether weight and drag, as well as mechanical stiffness and damping between nodes. Each tether segment is a mass-spring-damper system bounded by a node on each end. The nodes are placed along the tether length using cosine spacing, which allocates more nodes toward the aircraft than near the winch (see Figure 5). This is to reflect the increased tether bending downwind. Compared to linear node spacing, cosine spacing provides similar fidelity for lower computational cost. Most results presented involve a fixed tether length of 350 m with 15 nodes, thereby omitting the need for a ground winch. 350 m was chosen as this is the typical tether length at the start of reel out. The total number of states in this configuration is 102.

Figure 5 shows the primary coordinate system. The flat-Earth frame is described by the *X*, *Y*, and *Z* axes pointing north, east, and down, respectively, with the winch placed at the origin. The wind is assumed to be steady at 10 m/s and travels from south to north (i.e., in the positive *X* direction).

Figure 5: Coordinate systems and the numbering of tether nodes, showing N = 10.

A second frame, called the wind frame, is also shown in Figure 5 (not to be confused with the wind axis/body axis distinction in the aerodynamic sense). The main features are the  $Z_w$  axis pointing in the opposite direction to the wind, and the  $X_w$ - $Y_w$  pair forming a plane normal to the wind. The purpose of the wind frame is to determine a second set of Euler angles  $\phi_w$ ,  $\theta_w$ , and  $\psi_w$  relative to the  $X_w$ - $Y_w$  plane. These wind-frame Euler angles help visualise the turning motion relative to the wind more intuitively, as a typical circular trajectory tends to form a plane at an angle approaching 90 degrees to the wind.

In figures that show the flight trajectory with the aircraft visualised, such as Figure 6 and Figure 8a, the aircraft is drawn to scale. All 3D trajectories are drawn with respect to the Earth frame.

# 3 Static trim without gravity

'Static trim' refers to fixing the control surfaces at a constant deflection (as opposed to periodically forcing them, as discussed in subsequent sections). In an environment with no gravity and constant tether natural length, an aircraft trimmed for circular flight will converge to an orbit normal to the wind with the centre at [Y, Z] = [0,0] (i.e., overlapping the winch on the YZ plane). Figure 6 shows a few such trajectories, where the aircraft has been trimmed for  $6^{\circ}$  angle of attack  $\alpha$  and zero sideslip  $\beta$  at different radii. The radius is correlated to the roll angle  $\phi_w$  in the wind frame – henceforth referred to as the lean angle. For left turn cases as shown, negative  $\phi_w$  means more left roll, i.e., leaning more into the turn.




Figure 6: Circular trajectories without gravity for  $\alpha = 6^{\circ}$  and  $\beta = 0^{\circ}$ .

Figure 7 shows the relationship between  $\phi_w$  and R, tether tension T at the winch, and control surface deflections (aileron  $\delta_a$ , elevator  $\delta_e$ , and rudder  $\delta_r$ ). Trimming and stability were determined using the numerical continuation software AUTO 07-P (2021) interfaced in the MATLAB/Simulink environment via the Dynamical Systems Toolbox (Coetzee et al., 2010). Some notable features of Figure 7 are:

- Rolling is the primary mechanism for changing the radius. In (d), leaning more into the turn reduces the radius.
- In (e), there is a maximum for the tether tension at 2.8° lean out. Achieving higher tension is correlated with higher power for energy generation. A potential explanation for the optimal angle being slightly above 0° is that at 0°, the wing is normal to the wind and hence receives the most wind. However, the outer wing travels slightly faster than the inner wing and hence generates more lift. A few additional degrees of leaning out equalise the lift on both wings for optimal power generation. Beyond 2.8°, leaning out results in less pulling force.
- In (a-c), leaning in requires more left turn aileron, left rudder, and nose-up elevator. This is consistent with conventional piloting sense. However, at  $\phi_w = -22^\circ$ , the slope in the aileron diagram changes from negative to positive, which is accompanied by a stability change. Trajectories beyond this lean angle are statically unstable and involve static aileron trim in the opposite sense: more lean out (right roll) requires less right-roll aileron at trim. It should be noted that this behaviour does not mean control reversal as the aileron still maintains the conventional rolling power during transient motions; only the value at static trim is affected. This change of slope is not unexpected, as similar behaviour for elevator deflection has been observed in the pitch dynamics of statically unstable aircraft (Nguyen et al., 2022).


- Lean-out turns with zero sideslip are only achievable in tethered flights. In free flights, a coordinated right-roll turn cannot result in a left-turning trajectory as seen in Figure 6.

Figure 7: Relationship between lean angle and control surface deflections (a-c), radius (d), tether tension (e) to achieve trimmed flight at  $\alpha = 6^{\circ}$  and  $\beta = 0^{\circ}$ . The minimum radius is limited by rudder travel range (10°).

An aircraft trimmed for lean out without artificial stabilisation will diverge as shown in Figure 8. The aircraft in Figure 8a and all other trajectory visualisations are drawn 0.5 second apart – except for Figure 6 and Figure 15. Regarding the instability during lean-out in Figure 8a, due to a lack of restoring moment in roll, the radius increases further when there is a right-roll disturbance from trim. Further right roll causes a loss of lift in the positive-X direction, causing the nose to drop and eventually trajectory divergence. Given enough time, the aircraft converges to a right turn with low  $\alpha$  but high  $\beta$  – effectively swapping the two variables. As the control surface deflections have not changed, the trajectory is an uncoordinated right turn with right-

roll aileron and left rudder. The tether force is significantly lower in this flight regime as seen in Figure 8b. Therefore, high sideslip turns are inefficient for power generation (in addition to other conventional aeronautics issues associated with high sideslip flights).

Figure 8: Trajectory (a) and time history (b) of the lean-out instability at  $\phi_w = 15^\circ$ .

As the divergence involves coupling between roll and pitch, feedback stabilisation must incorporate the dynamics from both channels. The following feedback law was tested in Figure 9

$$\delta_a = \delta_{a_0} + K_1 (\phi_{w_0} - \phi_w) + K_2 (\theta_{w_0} - \theta_w)$$
 (1)




where  $\theta_w$  is the pitch angle in the wind frame. The subscript 0 denotes the value at trim.  $K_1 < 0$  and  $K_2 > 0$  are two proportional gains (note that  $K_1$  is negative due to sign convention in the roll channel: positive  $\delta_a$  gives negative  $\phi$ ). This feedback law stabilises the lean-out turn by cross-feeding both roll and pitch dynamics into the aileron. In particular, the second term in equation (1) provides an opposing aileron deflection when there is a disturbance in roll. The third term generates a positive (left roll) aileron when the pitch angle  $\theta_w$  drops below its trimmed value  $\theta_{w_0}$ , which indicates a widening turn with a dropping nose similar to in Figure 8a. Both the second and third terms are required for stabilisation, although cross-feeding other slow variables from the roll and pitch axes might achieve a similar stabilisation effect. Two sets of feedback gains are shown in Figure 9: appropriate gains providing stabilisation in Figure 9a and excessive gains, causing instability in Figure 9b.

Figure 9: Cross-feeding roll and pitch into aileron to stabilise a 15° lean-out orbit, showing responses to appropriate and excessive gains.

For verification, the results are now cross-checked with prior studies that utilised simplified point-mass models with no tether dynamics. Equation (2.10) in (Trevisi, 2024) approximates the relationship between the cone opening angle  $\Gamma$  and the tether force at the aircraft  $T_K$  for maximum power production (assuming no gravity). With reference to Figure 10a and ignoring the slight force misalignment with the cone geometry due to tether sag, this equation can be reinterpreted in our current coordinate system and notation as

$$\sin \Gamma = \frac{mu^2}{T_K R} \tag{2}$$

where m is the aircraft mass and u is the forward velocity in the body axis. Equation (2) can be improved by including a contribution from the tether weight  $m_T$ . (Trevisi et al., 2020) suggests lumping the aircraft mass with one-third of the tether weight  $m_T$ . This results in

$$\sin \Gamma = \frac{(m + m_T/3)u^2}{T_K R} \tag{3}$$

Figure 10: Cone geometry in gravity-free trimmed flight (a). Tether force at the winch vs. reference plane roll angle (b). Left- and right-hand terms of equation (3) vs. reference plane roll angle (c). Note the slight force misalignment in (a) due to tether sag.

Maximum power is achieved when the tether force T at the winch is the highest. This force is reached at  $\phi_{RP} = 2.8^{\circ}$  as shown in Figure 7e, which is reproduced in Figure 10b. In Figure 10c, the left- and right-hand sides of the approximated equation (3) are plotted. Equation (3) predicts an optimal reference plane roll angle of 0.63°. This is very close to the true value – despite the difference between the full and point-mass models – and provides additional confidence in the results shown thus far.

# 4 Dynamic trim without gravity


The discussed orbits above are all normal to the wind. Circular flights at different angles to the wind require adding energy to the system through periodic forcing of a control surface – henceforth referred to as 'dynamic trim'. Dynamic trim can be achieved through a simple feedback law

$$\delta_{a/e/r} = \delta_{a/e/r_0} + A\sin\psi_w \tag{4}$$

where  $\delta_{a/e/r}$  is one of the three control surfaces, A is a constant in degree and  $\psi_w$  is the heading angle in the wind plane.  $\psi_w$  = 0° indicate that the nose is pointing in the positive  $X_w$  direction, which is also the negative Z direction (i.e., nose vertically up).



Equation (4) is a way of deflecting a control surface periodically based on the aircraft's position in the orbit. The resulting trajectories are illustrated in Figure 11 for a forcing amplitude of  $A = 2^{\circ}$ . The control surface deflections at static trim are  $[\delta_{a_0}, \delta_{e_0}, \delta_{r_0}] = [-1.1^{\circ}, -6.2^{\circ}, 6.5^{\circ}]$ , which gives an orbit at 6° angle-of-attack, zero sideslip, and  $\phi_w = -37^{\circ}$ . In all dynamic trim cases in Figure 11, the circles are shifted from the original centre at static trim. This demonstrates the ability to fly against the wind during parts of the orbit. Aileron is the most effective control effector due to it shifting the circle centre from the static location by the largest distance (74.7 m for aileron, versus 72.3 m for elevator and 25.8 m for rudder). In all cases, the external forcing also causes periodic variation in the aircraft states and outputs, notably  $\alpha$  and  $\beta$ . Aileron forcing has the largest variation in both, although this can be attributed to a larger displacement of the centre, which requires more portions of the orbit to be against the wind, rather than the ineffectiveness of aileron. For the rest of this paper, aileron forcing is used.



Figure 11: Dynamic trim at 2° forcing amplitude using different control surfaces.

The effect of changing the forcing amplitude is shown in Figure 12. Larger *A* helps shift the circle further away from its static position, although this comes at a cost of larger variation in all state variables.

Another feature to note is that for different values of A, the circles are not simply shifted vertically down but with a slight offset in the positive Y direction (not clearly visible in Figure 12). This can be attributed to the phase lag between when  $\delta_a$  is

deflected and when a change in aircraft response takes effect. Although there is no known direct formula to calculate this continuous phase change, we can compensate for the phase lag by artificially shifting the reference point for  $\psi_w = 0^\circ$ 

$$\delta_a = \delta_{a_0} + A\sin(\psi_w + \Phi) \tag{1}$$

where  $\Phi$  is a constant indicating the amount of artificial phase shift. The effect of adding  $\Phi$  is shown in Figure 13, where  $\Phi = 189^{\circ}$  shifts the circle up with no offset in the *Y* direction. By combining *A* and  $\Phi$ , one can move the orbit to a different point in space (subjecting to physical constraints, such as control power, aerodynamic stall, and sideslip-induced instability).

Figure 12: Effect of changing the forcing amplitude A.

The oscillation in  $\alpha$  and  $\beta$  can be reduced by engaging the elevator and rudder. Using the following proportional feedback law

$$\delta_e = \delta_{e_0} + K_e(6 - \alpha)$$
  

$$\delta_r = \delta_{r_0} + K_r(0 - \beta)$$
(1)

where  $K_e < 0$  and  $K_r > 0$  are proportional gains (negative  $K_e$  due to sign convention), the oscillations can be reduced as shown in Figure 14. Furthermore, the circle centre is shifted up, indicating higher aerodynamic efficiency. It can be concluded that there are benefits to reducing the oscillation in  $\alpha$  and  $\beta$  during circular flights. Subsequent results will not consider the feedback law in equation (1) – the elevator and rudder are static, and only aileron forcing is used.

Figure 13: Artificial phase shift at  $A = 2^{\circ}$  to move the circle around the original static trim centre.

Figure 14: Increased 'shifting power' due to reduced  $\alpha$  and  $\beta$  oscillation provided by elevator and rudder compensations. Note that  $K_e = -2$ ,  $K_r = 4$ , and  $\Phi = 189^\circ$ .

An explanation of how dynamic trimming works is now provided. Figure 15a shows three example circular trajectories in static trim. The middle orbit is trimmed for zero sideslip at  $[\delta_{a_0}, \delta_{e_0}, \delta_{r_0}]$ , while the top and bottom ones experience an aileron offset of  $\mp A$  degrees. As expected, the top trajectory has a wider radius due to more negative (right roll) aileron trim, while the opposite is true for the bottom trajectory. The top trajectory pulls harder on the tether, leading to more elastic extension and hence flies further away from the winch (the opposite is true if a rigid stick is used to represent the tether: more lean out

leads to an orbit closer to the winch). Dynamic trimming using a forcing amplitude of A can be described as the aircraft traversing between the three static trim conditions. Referring to Figure 15b, when ψ<sub>w</sub> is at -90°, the aileron is at its minimum of (δ<sub>a0</sub> - A), while ψ<sub>w</sub> = 90° gives maximum aileron. Moving between these static trim conditions means periodically changing the radius at different parts of the orbit, causing the circle plane to be angled as seen in Figure 15b. This also creates an asymmetry in the aerodynamic force at ψ<sub>w</sub> = 90° and ψ<sub>w</sub> = -90°, which shifts the circle to the left during dynamic trimming
(not shown in Figure 15). The result is a circular trajectory at a non-90° angle to the wind. Dynamic trimming can also be described from a frequency response perspective. Using equation (4), the aileron undergoes harmonic forcing at one of the resonance frequencies (the rate of the aircraft completing one circle). This resonance adds enough energy to the system to enable flying in a circle that is not normal to the wind.

Figure 15: Top-down view of three generic static trim trajectories (left turning) and their periodic forcing components (a). Composite static trim diagram for illustrating the mechanism of dynamic trim (b). Only the middle trajectory in (a) is trimmed for zero sideslip.

# 5 Dynamic trim with gravity

Gravity adds an external downforce to both the aircraft and the tether. Consider the static trim case of  $[\delta_{a_0}, \delta_{e_0}, \delta_{r_0}] = [-1.1]$  or  $[-6.2]^\circ$ ,  $[-6.2]^\circ$ ,  $[-6.5]^\circ$ , giving  $[\phi_w] = -37$  (same configuration as in Figure 11). Figure 16a shows that adding gravity shifts the circle centre of static trim down and to the side, resulting in a plane of orbit that is no longer normal to the wind. The down shift is due to aircraft and tether weights, while the side shift is attributed to the difference in velocity (and hence lift) when the aircraft

is travelling up versus down. Both the down and side shifts cause the states and outputs to vary periodically, including  $\alpha$  and  $\beta$  as shown in Figure 16b.

Figure 16: Effect of gravity on static trim: trajectory (a) and time history (b). The wind direction in (a) is out of the page.

The loci of orbit centres for dynamic trim are also shifted in a similar manner. In Figure 17, the 'contour lines' for the same forcing amplitudes A no longer centre around the origin and become increasingly distorted with higher A, indicating increased nonlinearity due to gravity and crosswind. If we seek an orbit above ground level with no offset in the Y axis, then no such trajectory exists for  $A = 1^{\circ}$ . This suggests that to fly some above-ground orbits without violating a minimum height requirement, a certain level of periodic forcing is required.

Although the mechanism of circular flight has been explained, all results are still an academic study. The proposed dynamic trim law in equation (1) assumes prior knowledge of the static deflections at trim. Even when those trim values are known, such a feedback law cannot maintain a straight reel-out. Figure 18 shows the aircraft trajectory under aileron forcing as described by equation (1) with the winch active. The reel-out 'cylinder' is not straight, suggesting that in order to track a reference trajectory, the forcing amplitude and phase shift have to be constantly adjusted.


Figure 17: Loci of orbit centres (dashed lines) at different forcing amplitudes A in the presence of gravity. Two example orbits are drawn in solid lines. Figure shows both 2D (a) and 3D (b) projections.

Figure 18: Reel out using dynamic trim (aileron forcing – equation (1)).

#### 6 Conclusions


The dynamics of fixed-wing tethered aircraft in circular flight have been examined. Static trim analysis (no moving control surface) in a gravity-free environment shows that the lean angle determines the orbit radius, and the highest tether tension is achieved at a lean angle slightly above zero. Static stability may be lost when the lean angle exceeds a critical value, although artificial stabilisation can be done through cross-feeding pitch and roll into the aileron. In the presence of gravity, the orbit centre can be adjusted through periodic forcing of one of the control surfaces – most effectively the aileron. The basis of these ideas can be utilised to construct a flight control system for AWES.

#### Appendix: practical considerations for use with numerical continuation

This section outlines a few numerical challenges involving time simulation and numerical continuation of the airborne wind system considered. We also propose a few methods to remedy those issues.

Firstly, the heading angle  $\psi$  in the Earth frame tends to increase monotonically when the aircraft flies a circular orbit oriented close to 90° to the ground, such as in Figure 5. The resulting  $\psi$  time history is shown in the top half of Figure A1. A dynamical system with such a response is not considered periodic (repeating), and hence cannot be solved by the numerical continuation software AUTO. However, the sine and cosine of  $\psi$  is periodic. We can represent  $\psi$  using the two states  $s_{\psi}$  and  $c_{\psi}$  based on the Hopf normal form

$$\dot{s}_{\psi} = s_{\psi} + \dot{\psi}c_{\psi} - s_{\psi}(s_{\psi}^{2} + c_{\psi}^{2})$$

$$\dot{c}_{\psi} = c_{\psi} - \dot{\psi}s_{\psi} - c_{\psi}(s_{\psi}^{2} + c_{\psi}^{2})$$
(A1)

where the rate of change of  $\psi$  follows the definition from the literature

$$\dot{\psi} = q \frac{\sin \phi}{\cos \theta} + r \frac{\cos \phi}{\cos \theta} \tag{A2}$$

Figure A1: A typical time history of the heading angle during circular motion.

This interpretation of the Hopf normal form works because as long as  $s_{\psi}$  and  $c_{\psi}$  satisfy the physical constraint  $s_{\psi}^2 + c_{\psi}^2 = 1$ , equation (A1) reduces to

$$\dot{s}_{\psi} = \dot{\psi}c_{\psi} 
\dot{c}_{\psi} = -\dot{\psi}s_{\psi}$$
(A3)




which is identical to

$$\frac{d}{dt}(\sin\psi t) = \dot{\psi}\cos\omega t$$

$$\frac{d}{dt}(\cos\psi t) = -\dot{\psi}\sin\omega t$$
(A4)

It is worth noting that equation (A1) is numerically stable, whereas equation (A3) is not. This means that assuming the system with the original equation  $\dot{\psi}$  is stable, its representation using (A1) is also stable, as a perturbation to either state in (A1) will be damped out, bringing the trajectory back to the same attractor. This 'restoring force' is provided by the cancelled-out terms in (A1) when  $s_{\psi}^2 + c_{\psi}^2 = 1$ . If these terms are removed as in equation (A3), a perturbation to either state in (A3) will cause instability. The continuation software AUTO cannot solve for the system with (A3), but can work with (A1). In both cases, as long as the physical constraint  $s_{\psi}^2 + c_{\psi}^2 = 1$  is maintained, the trajectory will be identical to that of the original system. Figure A1b shows the dynamics of equation (A1) when the physical constraint is satisfied. From here, the heading angle wrapped between  $-\pi/2$  and  $+\pi/2$  can be reconstructed from the MATLAB's *atan2* function – shown in the lower half of Figure A1a.

In gravity-free analyses, such as in Figure 6, the roll angle  $\phi$  may also increase monotonically in a similar manner to Figure A1. The Hopf normal form can be used again, replacing  $\dot{\psi}$  with  $\dot{\phi}$  to hide  $\phi$  in two states representing  $\sin \phi$  and  $\cos \phi$ .

The second issue is numerical stiffness. This is due to the pitch angle  $\theta$  in the Earth frame regularly approaching or crossing  $\pm 90^{\circ}$  during circular motion. From the definitions of  $\psi$  (see equation (A2)), as  $\theta$  approaches  $\pm 90^{\circ}$ ,  $1/\cos\theta$  approaches infinity. A similar problem is present in the roll angle  $\phi$ 

$$\dot{\phi} = p + q \frac{\tan \theta}{\sin \phi} + r \frac{\tan \theta}{\cos \phi} \tag{A5}$$

which tends to infinity as tan θ approaches 90°. The consequence is a very rapid change of direction in the Euler angle states.

Figure A2 illustrates the issue by comparing two circular orbits with the same tether lengths but different radii. The solid-line trajectory has a minimum pitch angle of –88.9°, which is only 4.9° less than that of the dashed-line trajectory. Although the


difference is small, its impact on the numerical solver is evident. This leads to a stiff system that requires small step sizes, sometimes enough to fail the continuation solver.

Figure A2. Euler angles of two circular orbits with  $\theta$  and  $\phi$  close to  $\pm 90^{\circ}$ .

A possible remedy is to use vertical instead of horizontal wind in gravity-free cases. The resulting trajectories are identical but at 90 degrees to each other. With vertical wind, the Earth frame coincides with the wind frame, resulting in the states  $\phi$  and  $\theta$  admitting equilibrium (static trim) or small-amplitude oscillations close to zero (dynamic trim). This significantly reduces the computation cost for both time integration and continuation.

Code and data availability: the simulation model is proprietary and is therefore not available for the public.

Author contributions: DN: funding acquisition, analysis, writing (original draft). ML: funding acquisition, software, writing (review and editing). EO: software, methodology, validation, writing (review and editing).

**Competing interests:** the authors declare that they have no conflict of interest.

# Acknowledgement

Duc H. Nguyen and Mark H. Lowenberg are supported by the UK Engineering and Physical Sciences Research Council (EPSRC), grant number EP/Y014545/1. The valuable insights and suggestions from other members of the Kitemill team are much appreciated. We also thank Dr Filippo Trevisi (Polytechnic University of Milan) for the discussion surrounding Figure 10.

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
