# Peer review of "Trimming a fixed-wing airborne wind system for coordinated circular flights"

_Wind Energy Science, 2025_

## Author Comment (AC1)

**RESPONSE REVIEWER 1**

**When designing a flight controller, insight into the trim conditions is very useful because it allows for designing a feed-forward controller, which then still needs to be combined with a feedback controller.**

This is an important point. We have incorporated the feedforward idea into the introduction on lines 46-47, reproduced below:

'Results are fundamental in nature and may not constitute a feasible feedback strategy. However, the idea can be expanded for a future flight control system with a feedforward term.'

**A weak point is that it is unclear if this finding can be generalized to other wings.**
**…**
**A weakness is the reliance on proprietary software to simulate the tethered wing. This makes it difficult to reproduce the results.**

Theoretical studies by Trevisi have produced remarkably similar results. From his PhD thesis (Conceptual design of windplanes, Polytechnic University of Milan, 2024), it was noted that: 'the ailerons are actuated cyclically to control the roll angle. The cyclic control of the roll angle redirects the lift to compensate gravity and thus to stay airborne.' Since our numerical study also concluded that the required control input must be periodic, it is likely that other tethered vehicles with similar configurations will exhibit similar behaviours. This similarity is now mentioned on lines 214-216 of the revised paper.

**The conclusion that the aileron is the most effective control surface is supported by one specific set of numbers. But it is not clear, for example, if the areas of the aileron and the elevator are identical. So it is unclear if this conclusion is specific to the one airplane investigated or under what conditions this statement is valid. This must be improved.**

We agree that the original supporting evidence for using aileron forcing was not convincing. An extended discussion is now provided on lines 212-219 of the revision, reproduced below:

'Whilst all three control effectors could theoretically be used for dynamic trim, it is recommended to use aileron for several reasons. Firstly, aileron forcing frees up elevator and rudder for pitch and yaw control, respectively. Trevisi (2024) also remarked that actuating the aileron cyclically is a direct way to change the direction of the lift vector from the main wing, thereby providing an effective means to navigate a tethered aerial vehicle in 3D space. This point is supported by a direct comparison between rudder and aileron forcing by Nguyen et al., (under review), which shows that aileron forcing provides superior power generation capability. Lastly, the circular trajectory of aileron forcing in Fig. 11 is shifted from the static trim location by the largest distance (74.7 m for aileron, 72.3 m for elevator, and 25.8 m for rudder), although the contribution of the large moment arm generated by the long wingspan should be noted.'

**The naming of the reference frames could be improved. Instead of "primary coordinate system," I would call it a NED (north-east-down) reference frame, which is more common in the literature and**

**more specific. Furthermore, it is not mentioned whether the "wind frame" is an inertial frame or not.**

The term 'primary coordinate system' has been removed from the revision, and the section describing the coordinate system has been rewritten (lines 93-104). The wind frame is an inertial frame – this point is now stated in the revision.

We have initially considered using $[N, E, D]$ instead of $[X, Y, Z]$ to denote the aircraft's location on the flat-Earth plane. However, moving away from the $XYZ$ sequence weakens the link between the coordinate orientation and its Euler angles $[\phi, \theta, \psi]$. Since the Euler angles on both reference frames (Earth and wind) play a crucial role in the paper's main argument, we want to make this link clear. The original notation is therefore retained: $[X, Y, Z]$ and $[\phi, \theta, \psi]$ for the aircraft's location and Euler angle on the flat-Earth plane, and $[X_w, Y_w, Z_w]$ and $[\phi_w, \theta_w, \psi_w]$ for their counterparts on the wind frame. To make the distinction between two coordinate systems clearer, we have split figure 5 into two subfigures, reproduced below.

[Figure]

Figure 5. (a) Earth-axis and (b) wind-axis coordinate systems. Both figures show a tether with 10 nodes.

**The formatting of the reference makes them hard to read: No bold, no cursive, no indentation, no clickable links to the DOIs.**

We have made all DOI links clickable. The formatting of reference section was indeed uniform and can be hard to read, but this is the style specified by the journal.

**Modelling with gravity was included, but not Included was a simulation where the wing is always flying above the ground.**

**...**

**It is not described how to fly the wing at a height larger than zero. Not even a statement, if this is possible or not using pure feed-forward control can be found.**

Although flying at a positive height is possible, this point was not made clear in the original submission. Fig. 18 shows the aircraft flying above the ground in a full reel-out cycle. The revised text discussing this figure (lines 285-294) now specifies the necessary conditions to fly at positive height: setting $A \geq 2^\circ$ and $\Phi$ around $220^\circ$. A discussion on trajectory observed is also included in the section.

**RESPONSE TO REVIEWER 2**

**The study of the dynamics of tethered aircraft has had some contributions, mainly within the AWE community, e.g. Loyd, Argatov, Milanese, Schmehl, and co-authors, (which might be worth citing). Nevertheless, many open equations still remain. Control is among the most studied topics in AWE (e.g [R1] and references therein), it remains one of the most critical aspects for a safe and reliable AWE system.**
**...**
**The fact that just circular paths are addressed should not be seen as a limitation. The study [R2] proposes "motion primitives" for tethered aircraft that are defined to be circular paths, but can compose virtually any continuous motion on the surface of a sphere (defined by the tether length) by concatenating these primitives.**

Thank you for your positive feedback. We have incorporated some of your points into the revised introduction in three instances:

Line 20: citing the original Loyd paper.

Lines 26-29: 'Since AWES lack a solid structural foundation seen in conventional wind turbines, the number of degrees of freedom expands from 1 to 6 (excluding structural deformation). Control, therefore, becomes one of the most critical aspects for safe and reliable AWES operations (Vermillion et al., 2021).'

Lines 40-47 now cite additional papers on novel control strategies for AWES. The revised text also classifies those papers based on the type of controllers. A more detailed discussion on these different feedback strategies will be presented in a future publication, where we propose a new control algorithm for rigid-wing AWES in circular flights.

**MINOR ISSUES**

**All equations in page 12 are labelled as equation (1).**

Fixed

**Table 1: Boxy-axis angular rates -> Body-axis ...**

Fixed

**RESPONSE TO EDITOR**

**Abbreviate "Figure" as "Fig.", except at the start of a sentence. Same for "Figures" which should be "Figs."**

Fixed.

**Similarly for "Equation" which should be "Eq.", or "Equations" which should be "Eqs."**

Fixed.

**Note that Fig. or Eq. should start with a capital when referring to a numbered equation.**

Fixed.

**There are some instances where you write "Figure X and Figure Y" which looks better when using the plural form, "Figs. X and Y". See lines 61, 96, 137.**

Fixed

**Citations should use proper bracketing. Use "(author, year)" when citation is not part of the sentence or "author (year)" when citation is part of the sentence. As an example look at line 165, which should read "Equation (2.10) in Trevisi (2024) approximates …".**

Fixed

**You omit any punctuation for equations . Please add proper punctuation, as also display style equations are part of the sentence.**

Fixed.

**ADDITIONAL CHANGES**

The term 'fixed wing' has been changed to 'rigid wing', including in the title. We believe the original term is more appropriate for comparison against rotorcraft, whereas the revised term aligns more with AWES literature (rigid wing vs. softkites).

Fig 10: $\phi_{RP}$ is relabelled to $\phi_w$ to keep the notations consistent. The discussion surrounding cone angle $\Gamma$ has also been revised to improve readability (lines 190-194).

Fig 15: the horizontal axis on the two trim diagrams was mislabelled. It should have read 'Z (m)' instead of 'Y (m)'. This has been fixed in the revision.

Eqs. (2) and (A4) in the original submission had typos, which have been fixed.